# The Effect of the Recycling Process on the Performance of Thermoplastic Vulcanizates Containing Recycled Rubber from End-of-Life Tires

**DOI:** 10.3390/polym17222992

**Published:** 2025-11-11

**Authors:** Maialen Narvaez-Fagoaga, Marina M. Escrivá, Zenen Zepeda-Rodríguez, Laura Diñeiro, Fernando M. Salamanca, Ángel Marcos-Fernández, Juan L. Valentín

**Affiliations:** Institute of Polymer Science and Technology (ICTP-CSIC), Juan de la Cierva 3, 28006 Madrid, Spain; mmontero@ictp.csic.es (M.M.E.); zenen@ictp.csic.es (Z.Z.-R.); ldineiro@ictp.csic.es (L.D.); fms@ictp.csic.es (F.M.S.); amarcos@ictp.csic.es (Á.M.-F.)

**Keywords:** elastomers, sustainability, recycling, thermoplastic vulcanizates, end-of-life tires

## Abstract

End-of-life tires (ELTs) are an important source of energy and materials, with ELT powder (ELTp) being a secondary raw material of increasing industrial interest. However, the complex structure and composition of ELTp rubber pose technological difficulties and scientific challenges in some high-performance applications in the rubber industry. The mechanical recycling of ELTp produces ground tire rubber (GTR) powder, which is used, among other applications in the rubber field, to prepare thermoplastic vulcanizates (TPVs) due to the interest in these materials in the automotive and construction sectors. Over the last few decades, different approaches have been explored to minimize the limitations of these TPVs, including their large particle size and poor compatibility with GTR powder in other polymer matrices. This study applies different recycling procedures to GTR powder, based on thermal, chemical and mechanical methods, and combinations thereof, to minimize interfacial issues with other matrices used in TPV preparation. The effect of the different rubber recycling processes on the performance of the resulting TPVs was evaluated, optimizing the fraction of recycled rubber from ELTp and the vulcanization system to enhance the mechanical properties and obtain industrially competitive products.

## 1. Introduction

The growth in the number of end-of-life tires (ELTs) represents a significant environmental challenge. It is estimated that one billion tires are discarded every year worldwide, constituting a considerable volume of waste that is challenging to degrade due to its composition [1]. Inadequate management of this waste has been demonstrated to have a significant impact on soil, water, emissions of volatile compounds, and the generation of microplastics [2]. In contrast, the recycling of ELTs has been demonstrated to have a considerable effect on the reduction of greenhouse gas emissions. Products manufactured from these tires exhibit a distinctly lower carbon footprint than those made from alternative virgin materials [3].

Tires are mainly composed of vulcanized rubber, steel, and textile fibers. After removal, mechanical processes such as shredding, metal/textile separation, and grinding are carried out to obtain recycled rubber in the form of crushed granules or powder [2,4]. The compositional complexity of this material poses significant challenges in its incorporation into high-value products. Consequently, various recovery routes have been developed, such as pyrolysis, devulcanization, and utilization as a filler in polymer matrices [5]. However, most of these routes are focused on applications with minor technical requirements (such as fillers for pavements or sports surfaces) or on energy recovery, which results in a deficiency in the fabrication of high-performance composites. The management of end-of-life tires (ELTs) is focused on low-level recovery methods within the waste classification system. These measures include energy recovery, often through incineration as a fuel derivative in cement kilns, and mechanical recycling to obtain ground rubber for low-value-added applications, such as civil engineering projects or playground surfaces. While these routes are standard, they present a challenge to the circular economy, as they do not recover high-value materials that would restore the original functionality of the rubber [6].

A promising valorization of ELT is its incorporation into thermoplastic elastomers (TPEs), especially thermoplastic vulcanized elastomers (TPVs) [1]. Thermoplastic elastomers are a class of materials that combine the elasticity of rubber with the processing properties of thermoplastics. These properties render them as a competitive alternative to recycled rubbers, particularly in the context of thermoplastic vulcanized elastomers (TPV). Vulcanized thermoplastics are a subcategory of thermoplastic elastomers (TPEs), in which the elastomeric phase is crosslinked by vulcanization, resulting in the formation of a rubber phase that is dispersed within a soft thermoplastic matrix [7]. During the vulcanization process, the rubber particles formed inside the thermoplastic matrix are cross-linked and form a rubber network. Once this structure is formed, the cross-linked rubber particles increase the elasticity of the formed TPV, prevent the formation of larger rubber aggregates, and thereby help to maintain the desired mechanical properties. These materials have a wide range of applications in various sectors. These sectors include, but are not limited to, the automotive industry, construction, sealant component production, profile, and gasket manufacturing [2,8,9,10].

The integration of this recycled material into new materials can be affected by various factors that present technical challenges, such as interfacial compatibility with polymer matrices, homogeneous dispersion, appearance of impurities, deterioration of thermal and mechanical properties, stability during processing, and the effects of aging. In addition, vulcanized rubber exhibits a three-dimensional crosslinked network with significantly limited chain mobility, which restricts molecular entanglement with other phases. Another significant aspect to be considered is the heterogeneity of the waste, which is manifested in the physical characteristics of the material, such as particle size, carbon black and ash composition, and variability in the degree of rubber cross-linking. These factors contribute to the complexity of reproducing the final properties of the material. Recent research has indicated that proper characterization of GTR (microstructure, elemental composition, and particle size distribution) is imperative for optimizing its application in composites [11]. Consequently, within the domain of research, there is an imperative to investigate diverse methodologies for treating ELT, with the objective of enhancing compatibility between disparate matrices and increasing the final properties of the resulting materials [12]. The aim of this project is to introduce powder from end-of-life tires into vulcanized thermoplastic elastomer (TPV) formulations. In order to determine the effect of varying loading ratios on the final material’s properties, the present study will evaluate a range of loading ratios. The impact of processing conditions on the mechanical properties of the material, as well as on the reprocessability of the final material, will be examined. Furthermore, the impact of these modifications on the microstructure of the material will be evaluated, with particular attention paid to the distribution of recycled rubber, the dimensions and configuration of the elastomeric phases, and the quality of adhesion at the interface between the rubber and the thermoplastic matrix.

## 2. Materials and Methods

### 2.1. Materials

The preparation of thermoplastic vulcanized (TPV) materials was achieved by utilizing high-density polyethylene (HDPE) supplied by Repsol as the thermoplastic matrix, in conjunction with ethylene-propylene-diene monomer (EPDM) supplied by Versalis (S. Donato, Italy) under the trade name of Dutral^®^ TER 4038 EP, which served as the elastomeric phase.

The utilization of end-of-life tire derivatives (ELTs) as complementary materials was a key element of the study. The first was end-of-life tire powder (ELTp) with a particle size of 800 μm, obtained by shredding and granulating discarded tires, kindly supplied by Valoriza Servicios Medioambientales (Madrid, Spain). The second material, designated as ELTp-B, is a modified commercial product consisting of a blend of ELTp and bitumen, together with specific additives for use in asphalt blends. This material was kindly provided by Cirtec (Madrid, Spain) as an ad hoc modification of the commercially available RAR-X^TM^ product, which does not incorporate additional inorganic ingredients.

Furthermore, a variety of solvents were employed, including acetone (Carlo Erba—Milan, Italy), toluene (J.T. Baker—Phillipsburg, NJ, USA), hexane-1-thiol, and n-hexylamine (TCI Chemicals—Eschborn, Germany). All chemicals were used in the same conditions as received unless it is explicitly indicated in the following sections.

### 2.2. Methods

#### 2.2.1. Reclaiming and Devulcanization Processes

ELTp and ELTp-B raw materials were subjected two types of treatment to improve the final properties of the TPV and achieve better interaction between the raw material and the EPDM rubber: first, a thermomechanical devulcanization process was carried out using an extruder (Haake Rheocord 9000, Thermo Electron Corporation—Waltham, MA, USA) at 180 °C and at a rate of 100 revolutions per minute (rpm) for 5 min (Figure 1a). The samples produced were labeled “Desv”. Secondly, a mechanical regeneration treatment was applied, for which an external mixer or two-roll mill (Comerio Ercole—Busto Arsizio, Italy) with rollers measuring 15 cm in diameter and 30 cm in length, operating at a friction ratio of 1:1.4. The raw material was then subjected to shear stress for 10 min, producing a strip of material that was easier to incorporate in the final blend (Figure 1b). Materials that undergo this process were designated as “Rod”. A treatment combining both processes was finally administered, and the samples were named “Desv Rod” (Figure 1c).

#### 2.2.2. Solid–Liquid Extraction and Chemical Probe Treatment

The ELT materials, both treated and untreated, were subjected to an extraction procedure to quantify the extractable compounds present in them and thus evaluate the effectiveness of the applied reclaiming and devulcanization treatments. In order to avoid the loss of powder during the extraction process, the samples were contained in cylinders made of stainless-steel wire mesh (mesh size of 180 μm) with a height and diameter of 2.0 cm. The cylinders were then filled with 1 g of the sample.

In order to eliminate soluble organic compounds from the samples, the cylinders were placed in centrifuge tubes and filled with acetone for a period of 24 h. After that time, the cylinders were removed, and the solvent was replaced by fresh acetone to continue with the extraction process for an additional 24 h. Finally, the cylinders were taken out of the solvent and allowed to dry prior to being weighed.

The process continued with the immersion of the acetone-extracted samples (contained in the cylinders) in toluene for a period of 48 h, renewing after 24 h with fresh solvent. This procedure extracts the non-crosslinked polymer chains (sol fraction) and also swells the rubber particles in order to facilitate treatment with a chemical probe.

Finally, the pre-swollen samples were immersed in a solution of 2 M hexanethiol and 4 M n-hexylamine in toluene for a period of 48 h at room temperature to cleave the poly- and di-sulfidic bonds in the networks [13]. After this treatment, the samples were cleaned using toluene for a period of 48 h, refreshing the solvent every 24 h, with the aim of ensuring the complete removal of residues from the solution. Finally, the samples were dried and their weight was measured.

#### 2.2.3. TPV Blends

The following ingredients were used to formulate a thermoplastic elastomer containing a matrix of high-density polyethylene (HDPE) and an ethylene propylene diene monomer (EPDM), as shown in Table 1. The concentration of each ingredient in the recipe was defined in parts per hundred of rubber (phr).

In order to evaluate the effect of ELT powder on the TPV, a fraction of ELTp equivalent to 20% and 40% of the total TPV was incorporated, replacing 30% and 60% of the original EPDM rubber in order to maintain the weight ratio between HDPE and rubber fraction at a constant value of 1.25. The remaining ingredients were adjusted in accordance with the amount of rubber, as illustrated in Table 2.

In those samples, ELTp and ELTp-B were used to study the effect of adding bitumen to the ELT powder (ELTp-B) on the mechanical properties of the prepared TPVs. Additionally, in order to evaluate the effect of different rubber recycling processes, the same recipes were used by replacing the raw ELTp and ELTp-B with those materials after regeneration and devulcanization treatments.

The effect of the cross-link density in the rubber fraction was also evaluated by increasing the sulfur concentration in the TPV. The sulfur concentration was increased from 2.74 to 5 phr, and the other vulcanization system ingredients were adjusted in accordance with the specifications provided in Table 3.

Finally, the tensile mechanical properties of the ELTp-B content were evaluated. In order to achieve this, the amount of ELTp-B with different treatments was reduced to 10% by weight of the total blend. Furthermore, the amount of the previous vulcanization system was maintained at a constant level, with the other ingredients adjusted as shown in Table 4.

#### 2.2.4. TPV Processing

This section delineates the procedure employed for the preparation and processing of mixtures intended for the manufacture of TPVs. The process is subdivided into three stages. Firstly, the elastomer compound was prepared using a two-roll mill. Secondly, this elastomer compound was added to the polyolefin to produce the TPV using an internal mixer. Finally, the press molding of the so-obtained TPV provides the samples for the mechanical analysis.

##### Rubber Compound

Initially, the rubber compounds were prepared in a two-roll mill. Initially, the mastication of EPDM allows the formation of a continuous rubber band. After that, ELTp or ELTp-B was added (except for the case of the virgin TPV that does not contain this ingredient) and mixed until optimal dispersion and homogenization were achieved. The subsequent ingredients were introduced in the following sequence: activators (ZnO and stearic acid), accelerators (TMTD and MBTS), and sulfur.

##### TPV Blend

The rubber compound performed in the previous step was incorporated into the HDPE matrix using an internal mixer (Haake Rheomix, Thermo Electron Corporation—Waltham, MA, USA) at a temperature of 160 °C and a rotor speed of 80 revolutions per minute (rpm). The internal mixer possesses a total capacity of 78 cm^3^ and is equipped with Bambury-type rotors. It is imperative to consider the filling factor, which reached 60%, equivalent to 42–44 g of blend, to ensure effective sharing and mixing effort.

The HDPE is first incorporated into the internal mixer and melted for two minutes. After that, the rubber compound (previously prepared) was incorporated until a total of 15 min of mixing was completed at a temperature of 160 °C.

##### Compression Molding

The TPV samples were press molded using a Collin 200P automatic press (Maitenbeth, Germany). The process was carried out at a temperature of 170 °C using a steel mold with dimensions of 10 mm × 10 mm × 1 mm. The compression molding process was carried out according to the following stages:The mold with the TPV sample was introduced into the press without pressure for 5 min.Subsequently, the pressure was increased until 50 bar, and it was maintained constant for a period of 10 min.The pressure was then increased until it reached 150 bar and maintained for 10 min.Finally, the sample was cooled up to 70 °C for a period of 10 min.

Once the TPV sheets were obtained, the specimens for the mechanical tests were obtained by using a CEAST pneumatic die cutter.

#### 2.2.5. Characterization Tests

This section presents the tests carried out for the characterization of the vulcanized thermoplastic elastomers in this work.

##### Mechanical Properties

A tensile test was carried out in order to evaluate the mechanical properties of the obtained materials. The tests were carried out in compliance with the ISO 37 standard [14], using the Instron 3366 universal testing machine. Type 2 dumb-bell test pieces were used, recording the average thicknesses at three different measurement points and establishing a test length of 20 mm to record the results by video. The test procedure was carried out at a speed of 500 mm/min, using a load cell of 1 kN. Five test pieces were analyzed, and the median value was taken as a result of each property: the stress at 50% strain or modulus 50 (M50), stress at 100% deformation or modulus 100 (M100), as well as the maximum stress and deformation at break.

##### Differential Scanning Calorimetry (DSC)

Differential scanning calorimetry (DSC) analysis was performed using a Netzsch DSC 214 (Selb, Germany) equipment to evaluate the HDPE crystallinity in the different TPV blends. Three temperature ramps were implemented with a nitrogen flow of 50 mL/min, at a heating rate of 10 °C/min. The first ramp eliminated the thermal history of the sample, covering a range from a temperature of −90 °C to 150 °C. The second ramp started at 150 °C, descending to −90 °C. Finally, the third ramp ended with an increase in temperature until it reached 150 °C.

##### Freezing-Point Depression

Additional differential scanning calorimetry analysis was performed to characterize the freezing-point depression of the solvent absorbed in the rubber compounds to estimate the formation of vacuoles and empty spaces at the rubber interface (with fillers or with ELP-p). Vulcanized rubber samples with an approximate size of 5 mm^3^ were swollen in cyclohexane, a solvent selected for its favorable behavior in terms of crystallization according to the DSC [15]. During the swelling process, the samples were protected from light to prevent photo-oxidative degradation [16]. The samples that reached equilibrium swelling were placed in a DSC capsule with an excess of solvent. In this test, a temperature ramp ranging from 25 °C to −40 °C has been programmed. The nitrogen flow rate was set at 30 mL/min, and the cooling rate was 5 °C/min.

## 3. Results

### 3.1. Characterization of ELTp and ELTp-B

The aim of this study is to evaluate the effect of different recycling procedures applied to ELTp to minimize the interfacial issues with other matrices used in TPV preparation. Initially, the mechanically recycled ELTp was treated by the company Cirtec with bitumen in a thermo-chemical treatment to obtain its commercial product, ELTp-B, for use in asphalt blends. Finally, two different recycling treatments were conducted on our laboratories on the ELTp and ELTp-B raw materials: a regeneration treatment and a devulcanization process. The first treatment was carried out on a two-roll mill, where shearing forces were applied to the material, thus breaking the rubber network randomly. On the other hand, the devulcanization treatment involves the use of an internal mixer in which the materials are subjected to high temperatures and shear forces, followed by post-treatment in a two-roll mill.

In order to characterize the effect of these recycling processes on the structure and composition of ELTp, a procedure based on solvent extraction and treatment with chemical probe was applied to all these materials derived from ELT-p. The results of this analysis are shown in Table 5. The treatment with acetone is able to extract the organic compounds in the studied samples, except for the soluble polymer. As is demonstrated in Table 5, ETLp-B contains a greater fraction of extractable organic compounds because of the presence of bitumen in the rubber powder.

The weight loss after extraction with toluene quantifies the extractable, non-crosslinked rubber chains in the samples, the so-called sol fraction. Table 5 shows that the sol fraction increases from the initial 2% observed in the raw ELTp to 6% after the devulcanization treatment. These results are related to the efficiency of the applied process to break the rubber network, demonstrating that the regeneration method is less efficient than the devulcanization process for both ELTp and ELTp-B. At this point, it is important to highlight the variation in the sol fraction produced by the thermo-chemical treatment with bitumen, reaching a value around 12% for the sample ELTp-B. This increase in the sol fraction clearly shows the efficiency of this industrial process to break down the rubber network. Nevertheless, these sol fraction results are the combination of two processes: the breaking of sulfur cross-links and the scission of rubber chains.

The ability of the applied recycling process to selectively break sulfur cross-links was evaluated by the selective cleavage of sulfur bonds by a chemical probe using a thiol-amine solution in toluene. The sol fraction obtained from ELTp after this treatment quantifies the amount of extractable rubber after the complete cleavage of poly- and di-sulfidic cross-links, which are the treatable bonds during a selective devulcanization process. According to this statement and the results shown in Table 5, the treatment with bitumen (sample ELTp-B) is able to increase the fraction of treatable (devulcanizable) sulfur cross-links compared with the pristine ELTp counterpart. On the other hand, when both ELTp and ELTp-B samples were submitted to a regeneration or devulcanization process, some fractions of these treatable sulfur crosslinks are actually broken, thus reducing the sol fraction extractable after the treatment with thiol-amine. In this sense, the obtained results prove that the applied devulcanization process is more selective than the regeneration treatment, although it is not able to achieve a complete breakdown of the treatable crosslinks.

### 3.2. Vulcanization Curves

The vulcanization curves of the EPDM rubber compounds were measured on a Rubber Process Analyzer (RPA 2000 from Alpha Technologies) at a temperature of 170 °C with a strain amplitude of 6.98% and a frequency of 1.667 Hz. The vulcanization process was studied on a Rubber Process Analyzer, RPA 2000, from Alpha Technologies (Wiltshire, UK) at a temperature of 160 °C with a strain amplitude of 6.98% at a frequency of 1.667 Hz. The time required to achieve 97% of vulcanization (t_97_) for the EPDM sample was 17 min (see Figure 2), supporting the successful vulcanization of the EPDM phase during the mixing process with the HDPE to achieve a TPV with appropriate mechanical properties.

Figure 2 shows that the incorporation of ELTp and ELTp-B to this EPDM compound drastically reduces the induction time and accelerates the cross-linking reaction. This behavior may be attributed to the strong effect of substances that contain the ELT rubber powder (e.g., processing oils, antioxidants, accelerants, and their by-products) in the re-vulcanization process. The rubber compounds filled with ELTp show a similar maximum torque as compared to the pristine EPDM sample, whereas the minimum torque is higher. The latter may be related to the increasing viscosity of the EPDM matrix by the addition of a solid vulcanized rubber particle. In contrast, the sample filled with ELTp-B shows a minimum torque value quite similar to that of the pristine EPDM compound with a lower maximum torque, because bitumen acts as a plasticizer in the compound, thus reducing the viscosity during the scorch time and the final stiffness of the rubber compound when the vulcanization process is completed.

### 3.3. Effect of ELTp Fraction on the Mechanical Properties of TPV

Vulcanized thermoplastics prepared in this work are based on a thermoplastic matrix of HDPE and a vulcanized rubber phase based on EPDM. Figure 3 shows the quite different tensile behavior of these individual polymer matrices as compared to the mechanical response obtained when they are blended in the TPV (shown in Figure 4).

It was described that the mechanical properties of TPV blends strongly depend on the dispersion of the elastomeric phase in the thermoplastic matrix [17], the rubber particle size [18], and the achieved morphology, which is influenced by the cross-linking of the rubber phase during the dynamic vulcanization [19]. Rubber particle size in the range of tens of micrometers is required to achieve enhanced ultimate tensile properties since the deformation mechanism of TPVs under tensile conditions is dominated by localized yielding of the semicrystalline thermoplastic matrix, and the rubber particles of smaller sizes seem to suppress the formation of interlamellar voids [18]. This seems to be the case for the prepared TPV blend, which achieves tensile strength exceeding 25 MPa and deformation at break exceeding 500%.

Incorporation of ELTp in the rubber phase resulted in a significant change in stress–strain curves as shown in Figure 4. TPVs containing 20% and 40% of ELTp show similar tensile stress behavior to the virgin TPV in the low deformation region, as is shown in the M50 and M100 values in Figure 5. This mechanical behavior is dominated by the semicrystalline thermoplastic matrix. For this reason, the effect of ELTp on the crystallinity of HDPE was evaluated by adjusting the melting peak enthalpy (ΔH_HDPE_) corresponding only to the thermoplastic fraction in the TPVs. The results reported in Table 6 suggest that the crystallinity of the thermoplastic matrix is not strongly affected by the incorporation of ELTp.

However, in those samples filled with ELTp, the premature failure of the thermoplastic matrix contributes to observing a remarkable drop in the ultimate tensile properties (see Figure 4). This behavior could be due to the presence of a high amount of ELTp with particle size in the range of hundreds of microns and a poor interaction between the ELTp and the EPDM matrix.

### 3.4. Effect of Recycling Procedures on ELTp

Regeneration and devulcanization treatments were applied to ELTp to break down the rubber network structure and increase the sol fraction in these recycled materials in order to minimize the interfacial issues between ELTp and the EPDM phase. Figure 6 shows the stress–strain curves and ultimate tensile properties of TPV samples with 20% and 40% ELTp that underwent the regeneration treatment (named ELTp rod), respectively. The regeneration treatment is not able to enhance the mechanical properties of the TPV that contains 40% of ELTp; however, the sample that contains 20% or regenerated ELTp is able to minimize the formation of interlamellar voids in the HDPE, preventing its premature failure and contributing to achieving the high elongation behavior typically observed in TPVs. In both cases, incorporation of regenerated ELTp rod does not produce a notable effect on the crystallinity of HDPE (see Table 6).

According to the results shown in Table 5, the partially devulcanized ELTp sample (named ELTp desv rod) contains a higher sol fraction and a lower fraction of remaining sulfur cross-links than the regenerated counterpart. It can be observed that both treatments, regeneration and devulcanization, are able to increase the deformation at break in those TPV samples with 20% ELTp, although there are no noticeable differences between them (see Figure 7).

### 3.5. Effect of Adding ELTp with Bitumen

The mechanically recycled ELTp was treated by the company Cirtec with bitumen in a thermo-chemical treatment to obtain its commercial product ELTp-B, that is able to increase the sol fraction of this product and activate it to promote the breakdown of the rubber network during the devulcanization and regeneration, as compared to the non-activated pristine ELTp (see data in Table 5). Figure 8 shows the stress–strain curves of TPVs filled with 20% ELTp and ELTp-B after being subjected to the regeneration and devulcanization treatments, respectively. A similar behavior was observed for both materials, although the tensile strength of regenerated and devulcanized ELTp-B is slightly higher than the ELTp counterparts (see Figure 8). This mechanical response may be related to two opposing mechanisms. On the one hand, ELTp-B samples contain a higher sol fraction, which may enhance the interactions between these recycled particles and the EPDM rubber phase. On the other hand, a significant proportion of the sulfur crosslinks are broken during the recycling processes. Although complete devulcanization cannot be achieved, the cross-link density and therefore the mechanical properties of this rubber powder are significantly modified. Additionally, ELTp-B samples contain a significant fraction of bitumen, which acts as a plasticizer, softening the final TPV.

### 3.6. Effect of Vulcanization System

Thermo-chemical treatment of ELTp with bitumen may improve the interaction between ELTp and rubber, as well as the efficiency of recycling treatments. When ELTp has been treated, the crosslinked polymer chains break down and become available again for vulcanization. Because of this, a portion of the vulcanization system introduced into the TPV formulations may be partially consumed to re-vulcanize the devulcanized ELTp-B and thus not be available for the cross-linking reaction with the less reactive EPDM rubber. The effect of adding ELTp-B on the vulcanization curve of the rubber phase is evident according to the results shown in Figure 2. For this reason, incorporation of an additional amount of sulfur would assist the crosslinking reaction of the EPDM phase in the TPV. Figure 9 shows the effect of adding 5 phr of sulfur on the stress–strain curve of TPVs filled with 20% ELTp-B subjected to a regeneration treatment (a) and a devulcanization treatment (b). Although the increase in the amount of sulfur slightly enhances the tensile strength of the TPV samples (Figure 9), this is not the main reason to explain the absence of the characteristic stress upturn at the high strain regimen (above 200%) for TPV when it is filled with recycled ELTp materials.

### 3.7. Effect of ELTp-B Fraction

An analysis of the behavior of ELTp with bitumen in thermoplastic vulcanizates (TPV), taking into account both the treatments applied and the increment in the vulcanization system, has been conducted. The findings of this analysis suggest a slight (not meaningful) enhancement in the mechanical properties of the material caused by the increment in the devulcanization degree and sol fraction of ELTp and the increment in the cross-link density in the rubber phase. Consequently, the disparity in the rubber particle size dispersed in the thermoplastic matrix seems to be the main factor that hinders the proper mechanical behavior of TPV at the high strain regimen. This difference can be explained by the comparison of ELT powder, which has a maximum particle size of 800 μm (defined as the 95-percentile) with a nominal particle size of 550 μm according to the information provided by the supplier, and EPDM droplets, which may have a diameter of around tens of micrometers according to the ultimate properties of HDPE/EPDM thermoplastic vulcanizates. A decrease in the fraction of ELT powder in the TPV reduces the number of high-sized rubber particles in the blend, enhancing the mechanical properties of TPV filled with these recycled ELTp-B products, as shown in Figure 10.

### 3.8. Evaluation of ELTp-EPDM Interface by Depression of the Freezing Point

In this section, the ELTp-EPDM interface will be studied by determining the freezing-point depression of a solvent imbibed in the rubber compound. This experimental approach has been previously used to obtain qualitative information on the structure of the rubber network [15] and to study filler-rubber interactions [16,21]. In the case of the unfilled EPDM rubber sample, two peaks, attributed to the freezing process of the solvent outside (higher freezing temperature above 0 °C) and inside (lower freezing temperature below −30 °C) the rubber matrix, are observed (see Figure 11). The anomalous low-temperature shift in peak, attributed to the solvent that is swelling the rubber network, is caused by the small mesh size of the polymer network that interferes with the nucleation process of cyclohexane. Consequently, the depression of the freezing point can be related to the cross-link density of the EPDM sample and, hence, the dimensional constraints that it causes on the solvent nucleation. Figure 11 shows that in the case of the ELTp sample, although it is possible to discriminate two freezing peaks, the freezing temperature of the solvent inside the rubber matrix is only slightly shifted towards lower temperatures (around −5 °C) as compared to the freezing peak attributed to the solvent excess that is in the DSC pan. This is caused by the decrease in the size restrictions inside the rubber compounds and should be related to a rubber network with lower cross-link density, or more probably, with the presence of larger empty spaces at the filler-rubber interface (with reduced dimensional constraints for solvent nucleation) in these ELTp samples.

Figure 12 shows the DSC freezing curves of cyclohexane in swollen EPDM filled with pristine ELTp and with devulcanized ELTp, respectively. In both cases, only one freezing peak is observed. It means that the solvent imbibed in these vulcanized rubber samples does not find any dimensional constraint to nucleate (as compared to the solvent in the DSC pan) because a considerable fraction of the solvent is in larger empty vacuoles, where the freezing process starts. This behavior is not modified by the devulcanization treatment of ELTp, indicating that, in both cases, there is a lack of interactions at the ELTp–EPDM interface.

### 3.9. TPV Reprocessing

Vulcanized thermoplastic materials can be reprocessed while preserving their original mechanical properties. Figure 13 shows that incorporation of ELTp and ELTp-B does not disturb the re-processability properties of these TPVs, maintaining their mechanical properties, thereby emphasizing their technological, economic, and environmental relevance.

## 4. Conclusions

Mechanical recycling of end-of-life tires provides a rubber powder (ELTp), which can be used as a secondary raw material in the rubber industry, including vulcanized thermoplastic elastomers (TPVs). The results obtained in this work provide relevant evidence of the challenges and difficulties of using ELTp as a filler in TPVs based on HDPE/EPDM blends, including higher viscosity, interference with the dynamic vulcanization process, poor compatibility with other rubber/thermoplastic matrices, and the large rubber particle size.

The particle size of the ELTp (in the range of hundreds of microns) is considerably larger than the EPDM droplets (in the range of decens of microns) dispersed in the thermoplastic matrix. This limits the mechanical performance of these TPVs when this secondary raw material is incorporated at elevated concentrations (higher than 20% *w*/*w*). Although the crystallinity of HDPE remains unaffected by the presence of ELTp, resulting in an almost unaltered mechanical response in the low strain regime (below 100% deformation), the premature failure of the thermoplastic matrix contributes to a remarkable drop in the ultimate tensile properties of TPVs.

In order to address these difficulties, different recycling procedures based on thermal, chemical, and mechanical methods, and combinations thereof, were applied to ELT-p. In particular, ELTp was treated with bitumen in a thermo-chemical treatment that demonstrated an efficient industrial process to break down the rubber network, increasing the rubber sol fraction to up to 12%. This commercially available material (named ELTp-B in this manuscript) is able to increase the fraction treatable (devulcanizable) sulfur cross-links as compared with the pristine ELTp counterpart, thus activating this material for further recycling processes. Finally, two different recycling treatments were conducted in our laboratories on the ELTp and ELTp-B raw materials: a regeneration treatment and a devulcanization process. It was proved that the applied devulcanization process is more selective than the regeneration treatment, although it is not able to achieve a complete breakdown of the treatable crosslinks.

The ELTp-based materials obtained after the application of these recycling treatments showed similar behavior regarding the mechanical properties of the TPV when they are used as fillers, resulting in a substantial improvement in the mechanical properties of the final product. In this sense, all TPV samples that were filled with 20% w/w of recycled ELTp are able to minimize the formation of interlamellar voids in the HDPE, preventing its premature failure and contributing to achieving the high elongation behavior typically observed in TPVs. It is important to note that the bitumen present in the ELTp-B-based materials acted as a plasticizer, thereby reducing the viscosity of those materials.

It was established that an increase in the quantity of vulcanizing agents (sulfur) has favorable effects on the mechanical properties of TPVs when devulcanized ELTp is applied as filler. Nevertheless, the fraction of treated ELTp incorporated in the rubber phase is the main factor that modifies (optimizes) the mechanical properties of TPVs.

This work demonstrates that treated ELTp is a viable and sustainable raw material to be used in the rubber phase in TPV formulations, contributing to the development of sustainable strategies for the recovery of tire waste, thus opening new perspectives for its integration into products of commercial interest.

## Figures and Tables

**Figure 1 polymers-17-02992-f001:**
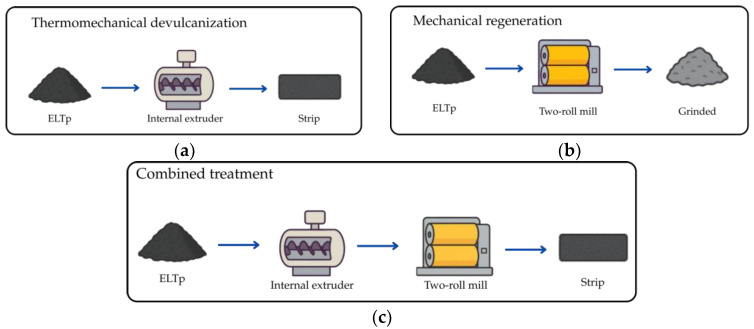
Representation of different treatment methods used in the project: (**a**) thermomechanical devulcanization, (**b**) mechanical regeneration and (**c**) combined treatment.

**Figure 2 polymers-17-02992-f002:**
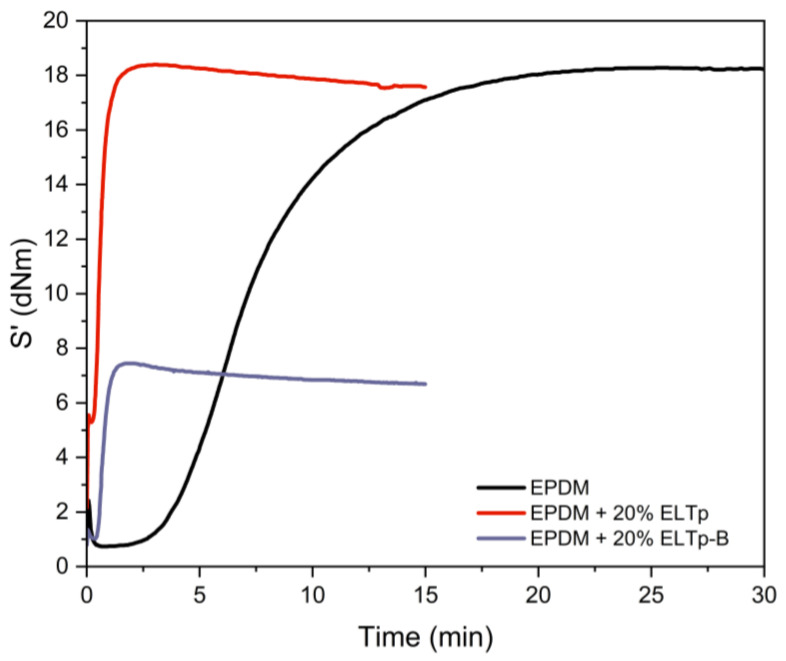
Vulcanization curves of the EPDM compound and the effect of filling with ELTp and ELTp-B, respectively.

**Figure 3 polymers-17-02992-f003:**
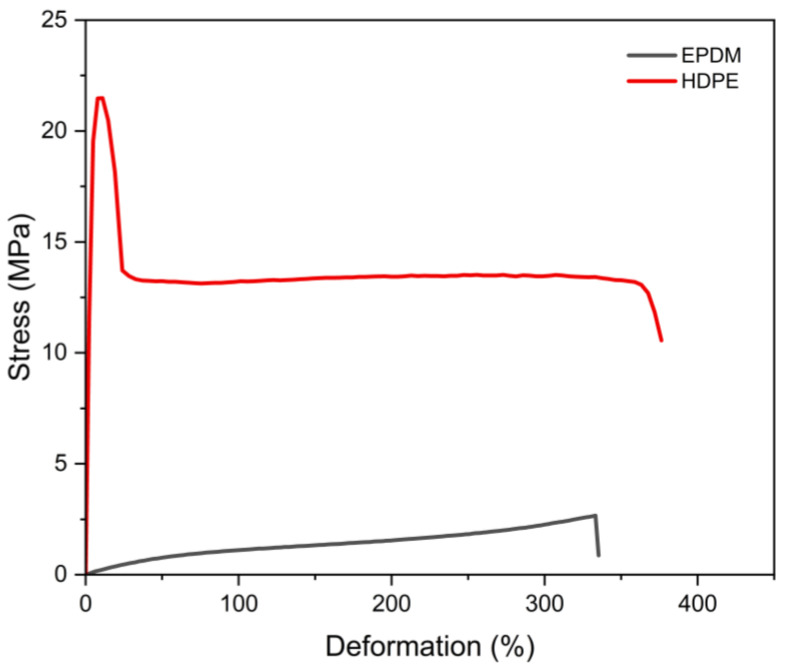
Stress–strain curves of thermoplastic HDPE and rubber EPDM materials used to prepare TPVs.

**Figure 4 polymers-17-02992-f004:**
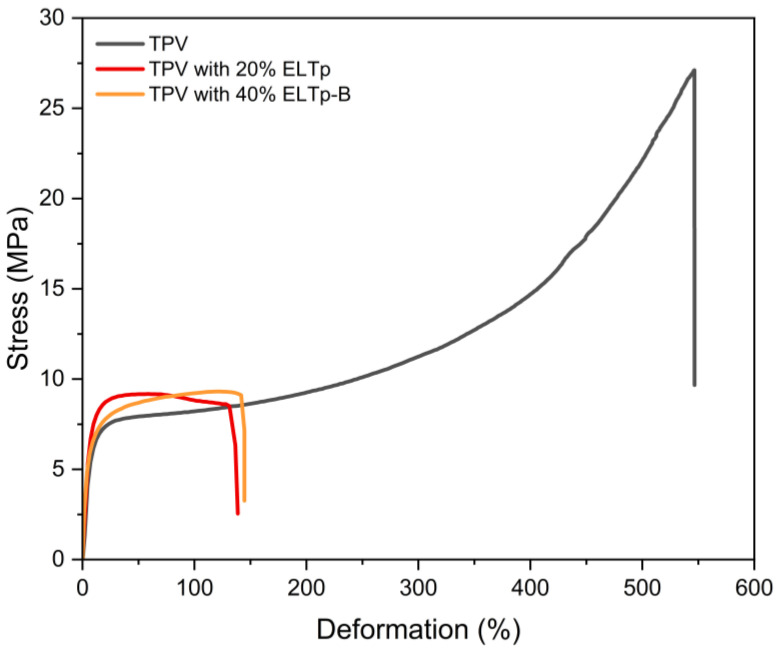
Stress–strain curves of TPV and TPV filled with different fractions of ELTp.

**Figure 5 polymers-17-02992-f005:**
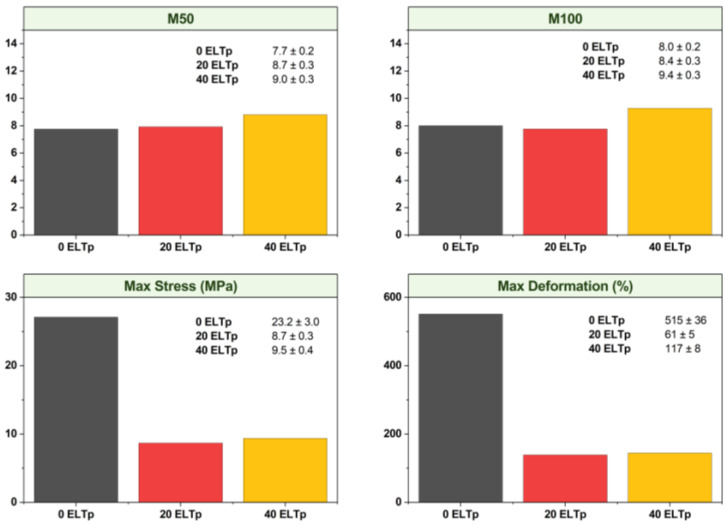
Mechanical properties of TPV with 0% (black), 20% (red), and 40% (yellow) ELTp.

**Figure 6 polymers-17-02992-f006:**
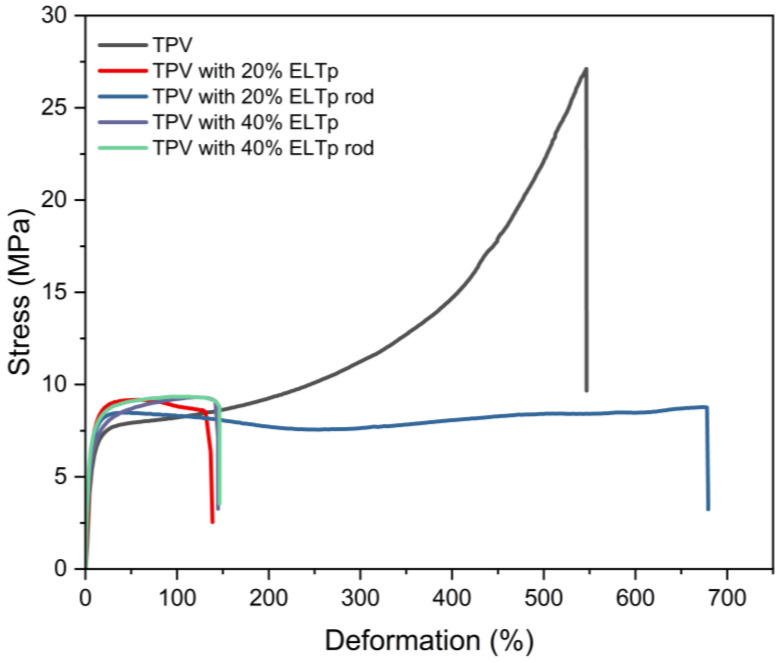
Stress–strain curves of TPV and TPV filled with different fractions of ELTp and regenerated ELTp rod.

**Figure 7 polymers-17-02992-f007:**
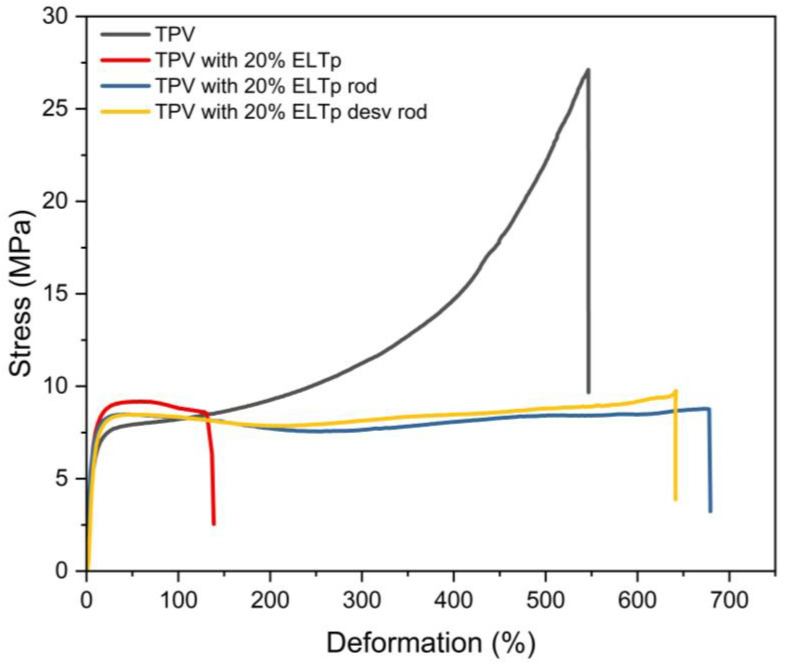
Stress–strain curves of TPV and TPV filled with 20% pristine ELTp, and after regeneration (ELTp rod) and devulcanization treatments (ELTp desv rod).

**Figure 8 polymers-17-02992-f008:**
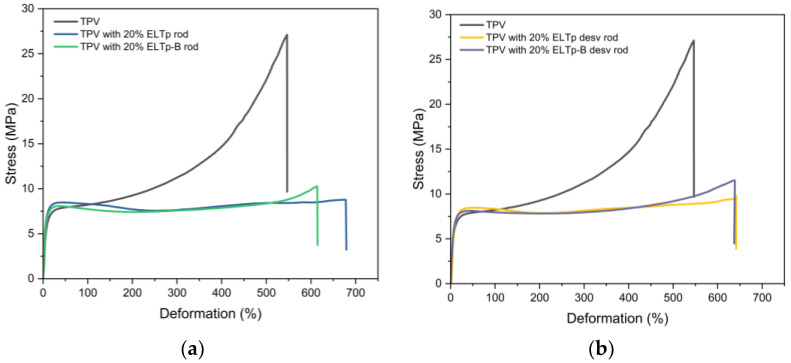
Stress–strain curves of TPV and TPV filled with 20% of (**a**) regenerated ELTp and ELTp-B and (**b**) devulcanized ELTp and ELTp-B.

**Figure 9 polymers-17-02992-f009:**
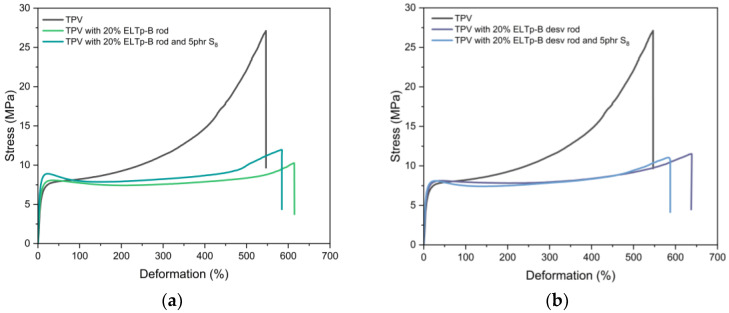
Effect of increasing the amount of sulfur on the stress–strain curves TPV blends filled with 20% of (**a**) regenerated and (**b**) devulcanized ELTp-B.

**Figure 10 polymers-17-02992-f010:**
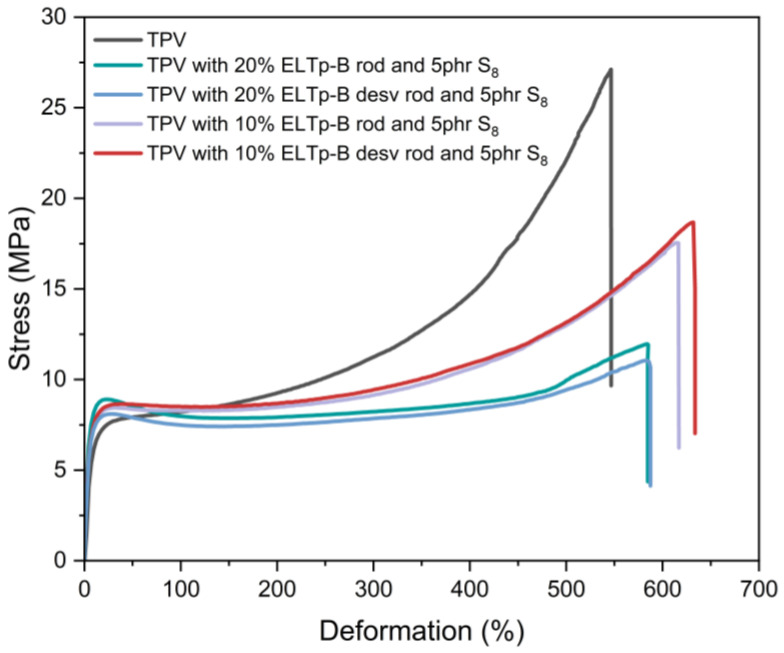
Stress–strain curves of TPV blends filled with a variable fraction of recycled (regenerated and devulcanized) ELTp-B and 5 phr of sulfur.

**Figure 11 polymers-17-02992-f011:**
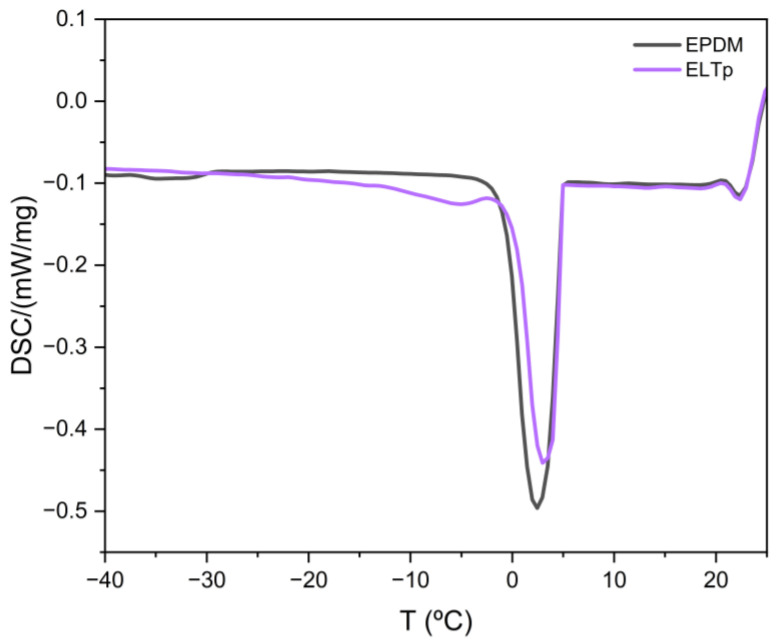
DSC freezing curves of cyclohexane in swollen EPDM and pristine ELTp.

**Figure 12 polymers-17-02992-f012:**
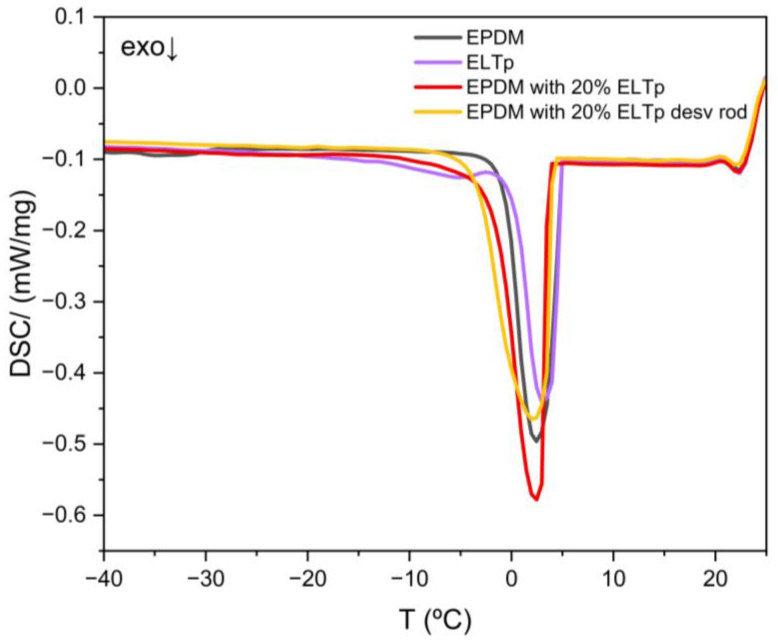
DSC freezing curves of cyclohexane in swollen EPDM samples filled with pristine ELTp and devulcanized ELTp.

**Figure 13 polymers-17-02992-f013:**
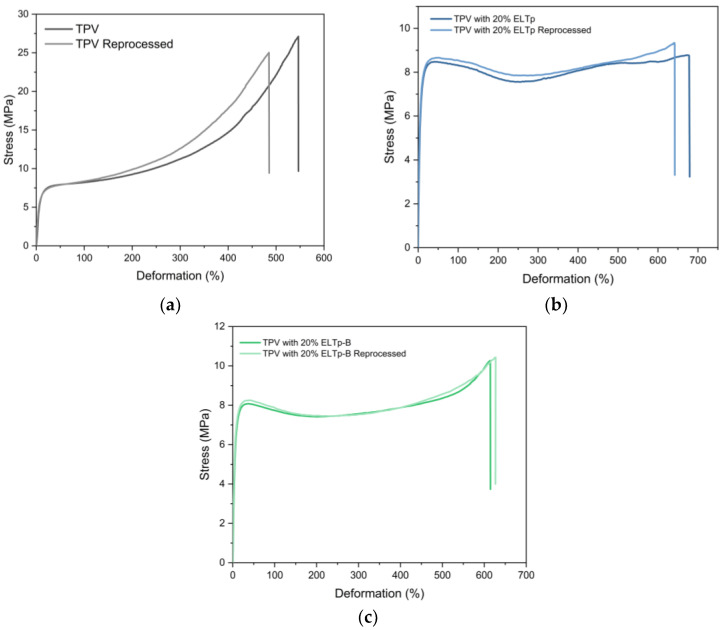
Stress–strain curves of reprocessed TPV. (**a**) TPV pre-processing without recycled material. (**b**) TPV pre-processing with 20% ELTp and regeneration treatment. (**c**) TPV pre-processing with 20% ELTp-B and regeneration treatment.

**Table 1 polymers-17-02992-t001:** Recipe of the virgin TPV.

Ingredient	Amount (phr)
HDPE	124.95
EPDM	100.00
ZnO	5.01
Stearic acid	1.02
TMTD ^1^	1.02
MBTS ^2^	0.48
S_8_	1.99

^1^ TMTD: Tetramethylthiuram disulfide. ^2^ MBTS: 2,2′-Dibenzothiazolyl disulfide.

**Table 2 polymers-17-02992-t002:** TPV blends with 20% and 40% ELTp/ELTp-B.

	20% ELT-p	40% ELT-p
Ingredient	phr	phr
HDPE	171.75	326.75
EPDM	100.00	100.00
ELTp/ELTp-B ^1^	72.04	310.61
ZnO	6.89	13.10
Stearic acid	1.41	2.68
TMTD	1.41	2.68
MBTS	0.67	1.27
Sulfur	2.74	5.21

^1^ Also including these materials after the regeneration and devulcanization treatments.

**Table 3 polymers-17-02992-t003:** Reformulation of TPV blends with 20% ELTp.

Ingredient	Amount (phr)
HDPE	171.75
EPDM	100.00
ELTp/ELTp-B	72.04
ZnO	6.89
Stearic acid	1.41
TMTD	2.57
MBTS	1.22
Sulfur	5.00

**Table 4 polymers-17-02992-t004:** TPV blends with 10% and 20% ELTp-B.

	10%	20%
Ingredient	Amount (phr)	Amount (phr)
HDPE	143.40	171.75
EPDM	100.00	100.00
ELTp-B	28.41	72.04
ZnO	5.75	6.89
Stearic acid	1.17	1.41
TMTD	2.57	2.57
MBTS	1.22	1.22
Sulfur	5.00	5.00

**Table 5 polymers-17-02992-t005:** Percentage of weight loss after solvent extraction and chemical probe treatment for ELTp and ELTp-B samples subjected to different recycling treatments.

	Initial		Regeneration		Devulcanization	
	ELTp	ELTp-B	ELTp Rod	ELTp-B Rod	ELTp Desv Rod	ELTp-B Desv Rod
Acetone	8.6	19.7	6.3	13.7	7.1	26.8
Toluene	2.0	11.8	3.1	10.4	6.4	14.6
Thiol-Toluene	19.0	35.2	16.0	19.5	7.8	13.5

**Table 6 polymers-17-02992-t006:** Crystallinity and enthalpy of HDPE in TPV compounds with 20% and 40% ELTp and ELTp rod, respectively.

Material	ΔH_tot_ (J/g)	ΔH_HDPE_ (J/g)	Crystallinity (%)
Theoretical HDPE [20]	293	293	100
Experimental HDPE	193.9	193.9	66.2
TPV	100.3	188.2	64.2
TPV with 20% ELTp	94.34	196.0	66.9
TPV with 40% ELTp	71.25	166.2	56.7
TPV with 20% ELTp rod	96.14	199.8	68.2
TPV with 40% ELTp rod	83.99	195.9	66.9

## Data Availability

The original contributions presented in this study are included in the article. Further inquiries can be directed to the corresponding authors.

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
