# Peer review of "The Effect of the Recycling Process on the Performance of Thermoplastic Vulcanizates Containing Recycled Rubber from End-of-Life Tires"

_polymers, 2025, doi:10.3390/polym17222992_

Round 1
Reviewer 1 Report
Comments and Suggestions for Authors
The manuscript reports "the effect of the recycling process on the performance of thermoplastic vulcanízates containing recycled rubber from end-of-life tires", it is an interesting topic, and work is well structured, however there are several issues that need to be corrected before continuing the publication process, following they are detailed:
-I recommend to improve the state of art of work, due less than 1 page is short.
-Please provide some properties of used materials such as HDPE, EPDM.
-It would be interesting to know what kind of additives have the ELT materials used in this work, I mean on depending of those additives the behavior of final material can varies.
-How established the proportion of HDPE and EPDM in TPV formulations?? And also for the rest of additives used in formulation.
Also please identify TMTD, MBTS, and 8.
-How decided that ELT powder on TPV must be 20 and 40%? And it is kind confusing that in table 2 use % and per to express content of materials in formulations.
-In line 152 and 250 there is a writing mistake.
-It would be interesting to insert a diagram to explain the treatment to ELTp and ELTp-B, of regeneration and devulcanization for better understanding. I mean, identify the structure and changes that happened with those treatments.
-How decide the conditions for evaluation of vulcanization curves??
-In lines 292-293, attribute the acceleration of cross linking reaction to the presence of additives in ELTp, but how is it possible to know the presence of those additives?
-I recommend to use colors that can be easy identificables for curves in figure 1, due the current colors are hard, for instance use red, green, blue, etc.
-In lines 325-326, indicate that the crystallinity of thermoplastic matrix is not affected by the incorporation of ELTp, however the crystallinity reported in table changed from 66 to 56%, so it is a significant change in this property. it is important to support this, may be including XRD analysis to support this asseveration.
-The premature failure of the thermoplastic matrix is attributed to the presence of particles of hundreds of microns or poor interaction between ELTp and elastomer matrix, according with theory which reason is more reasonable? I mean the authors need to be sure of justification of behavior of properties.
-The mechanical properties are reported in two figures, it is mandatory to avoid duplication of data so plenas delete one of the figures 2 and 3; same for 4 and 5; 6 and 7; 8 and 9;10 and 11; 12 and 13.
-Table 6 caption must indicate that is included data of melting enthalpy, not o only crystallinity.
-In DSC thermogram showed in figure 14, it is possible to observe a small peak around -5°C for ELTp, this signal is not discussed, please explain this. Please correct the scale on figure 14 and 15 (DSC thermogram) it must be from -40 to 20°C (left to right). Also please indicate in the heat flow axis Endo or Exo in axis, please. For DSC results reported in fig 14 and 15, is it not relevant the peak area? I mean this peaks represents the heat amount of this process, so the are is different for all samples reported, what is this represents?
-The conclusions need to be more concise, I mean, need to be shorted due are too long.
Reviewer 2 Report
Comments and Suggestions for Authors
The topic of tire recycling is very current and is becoming increasingly popular. Many problems are related to the fragmentation and separation of individual tire components, where, in addition to rubber, we are dealing with textile, steel, and aluminum cords. The work aligns with current research trends.
1. The title of the manuscript is consistent with its content. It is worth summarizing the main results of the research, its innovative potential, and the main conclusion.
2. The introduction to the work is somewhat modest, although it addresses many issues related to tire recycling. It would be worthwhile to expand it and highlight the novelty of the proposed research solution compared to existing solutions.
3. The literature review is very limited; I would recommend that the authors re-evaluate the state of the art in the field of current research. Despite this, the literature is relevant.
4. The broadcast program was developed correctly. I would like to draw attention to the possibility of introducing standards according to which the tests are performed or to expand the description of the research methodology to include a statistical analysis of the obtained results.
5. How were the components of the rubber pulp separated?
6. Blending parameters should be summarized in a table.
7. Line 250?
8. Fig. 8 - I would appreciate a more detailed description.
9. Both the figures and tables provide valuable information. It would be worthwhile to supplement the data with the number of repetitions (number of samples) and provide the measurement error, uncertainty, or standard deviation.
10. The final conclusions were very well developed.
Round 2
Reviewer 1 Report
Comments and Suggestions for Authors
After review the manuscript, the corrected version shows a significant improve compared with previous version, I wish to thank to the authors for consider most of the comments/observations done to the manuscript.
Author Response
Thank you for your suggestions.
Reviewer 2 Report
Comments and Suggestions for Authors
Thank you very much for sending the explanations and correcting the manuscript. The authors have completed the work and clarified all doubts. There is a minor problem in line 279. It needs to be corrected.
Author Response
Thank you for your suggestions. The errors located in line 179 and 279 were corrected.